# Coproduction of marine restoration with communities facilitates stronger outcomes

Richard K.F. Unsworth[1,2] (iD), Leanne C. Cullen-Unsworth[2], Emma Fox[2], Benjamin L.H. Jones[2], Flo Taylor[2], Sue Burton[3], Jetske Germing[4] and Ricardo Zanre[5]

[1]Swansea University – Singleton Park Campus, Swansea, UK; [2]Project Seagrass, Bridgend, UK; [3]Pembrokeshire Special Area Conservation, Wales, UK; [4]Pembrokeshire Coastal Forum, Pembroke Dock, UK and [5]WWF-UK, London, UK

## Case Study

**Keywords:**
blue carbon; biodiversity; restoration; marine; social science; engagement

**Corresponding author:**
Richard K.F. Unsworth;
Email: r.k.f.unsworth@swansea.ac.uk

## Abstract

Near-shore marine habitats are well-documented as diverse and productive social-ecological systems; their degradation and loss have led to growing interest in marine restoration. However, the literature offers limited consideration of the interactions between these projects and stakeholders and local communities. We present a case study showing how a stakeholder engagement strategy ultimately led to the co-production of a marine restoration project among scientists, stakeholders and local communities. Alongside biological recovery, we present the complex social, logistical and ecological lessons learned through this stakeholder engagement strategy. Principally, these relate to how the success of the project hinged on the point at which the project was co-developed with the input of local communities and strategic stakeholders, rather than in a disconnected, independent manner. This project demonstrates that for marine restoration to truly be successful, projects need to engage and work with local people from the outset, through open and early stakeholder engagement and particularly with the people possibly impacted by its presence. Projects need to be created not just for ecological design but also to be relevant and beneficial to a wide range of people. What we show here is that co-producing a project with communities and stakeholders can be complex but lead to long-term sustainability and support for the project, with strong ecological outcomes. To achieve this requires an open and flexible approach. Finally, this work showcases how the restoration of marine habitats can be achieved within a social-ecological system and lead to benefits for people and the planet.

## Impact statement

Effective marine restoration goes beyond ecological science, requiring meaningful stakeholder engagement and coproduction with local communities. Early and open collaboration with stakeholders ensures relevance and fosters long-term support for marine restoration. Marine restoration projects can align ecological goals with societal needs, creating benefits for both people and the planet. A flexible and inclusive approach to marine restoration projects is necessary to navigate social and logistical complexities.

## Introduction

Seagrass meadows are well recognised as social-ecological systems, supporting activities such as fisheries, boating, culture and recreation (Cullen-Unsworth et al., 2014; Foster et al., 2025). These habitats have experienced widespread degradation due to stressors like poor water quality, coastal development and unsustainable fishing (Dunic et al., 2021), necessitating increased restoration efforts in line with international and local legislation (Fu et al., 2024). Recent academic work has focused on advancing restoration techniques through ecological, engineering, electronic and biological innovations. However, restoration success is rarely assessed in relation to the local communities that use these areas. For restoration to be effective, the original stressors must be removed, and local support ensured (Unsworth et al., 2025). Embedding local communities at the core of restoration efforts is essential to establish and maintain low-impact, favourable conditions (Fox and Cundill, 2018). Despite evidence from terrestrial restoration showing that stakeholder engagement and co-production are crucial for success (Metcalf et al., 2015; Floyd et al., 2024), these social dynamics remain underexplored in marine, particularly seagrass, restoration projects. Project success for marine habitat restoration, such as seagrass, is predominantly documented in terms of planting outcomes rather than in positive outcomes for the site's coastal users, leading to reduced human impacts on the restored habitat (Unsworth et al., 2025). Here, we provide a case study of how a stakeholder engagement strategy ultimately led to the co-production of a marine restoration project in South Wales (UK) between scientists, local strategic stakeholders and communities. This was the United Kingdom's first major seagrass restoration project.

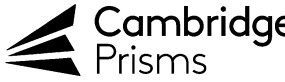



## Background

### *Dale Bay seagrass restoration*

Following a series of successful small-scale seagrass restoration trials within Dale Bay (also referred to as Dale Roads) in Pembrokeshire, West Wales (2017–2019), a restoration project was developed in 2019. This initiative was a collaboration between Swansea University, World Wildlife Fund (WWF), Sky Ocean Rescue and Project Seagrass. The Pembrokeshire Marine Special Area of Conservation (SAC) Officer and the Pembrokeshire Coastal Forum were brought in to aid project engagement. The project aimed to plant 2 ha of seagrass (*Zostera marina*) in a subtidal environment. Beyond the planting itself, the project sought to enhance understanding of seagrass meadows in the United Kingdom, showcase the potential of seagrass restoration to a wider audience and demonstrate how marine conservation can be achieved collaboratively with local communities. The project was designed around the "Bags of Seagrass Seeds Line (BoSSLine)" method, building on the success of initial trials and environmental assessments of the site (Unsworth et al., 2019). Dale Bay represents a rich social-ecological system, containing diverse habitats and high biodiversity. The site is regularly used by students from the nearby Dale Fort Field Studies Centre, and in 2012 the Welsh Government identified it (already a SAC) as a potential Highly Protected Marine Conservation Zone, a move that created a lot of local opposition. The bay also supports significant commercial and recreational activities, including fishing and sailing, while its sheltered waters provide a natural anchorage protected from prevailing south-westerly winds. Consequently, engaging stakeholders and securing support from the local community was recognised from the outset as essential for project success. Initial project site selection focused on areas with abundant chain moorings, with the idea of applying the BoSSLine method to spread seagrass among them. While moorings are often associated with "halo" patterns of seagrass loss, previous research has shown that seagrass can proliferate between moorings

where background environmental conditions are suitable. The project, therefore, aimed to promote this proliferation in largely bare seabed areas that were otherwise unthreatened by human impacts (Unsworth et al., 2018).

### *Understanding community views*

Initial validation of the project concept was undertaken through high-level "key informant" engagement, involving stakeholders and local individuals with experience in marine conservation and science (Supplementary Materials 1 and 2). This process successfully secured approval from the local Yacht Club and other key stakeholders to use the seabed and obtained the necessary regulatory licence to carry out the restoration. On paper, the project was fully authorised and legally permitted to operate. Following this, the project expanded its engagement to the broader community (Figure 1), conducting targeted public meetings, one-to-one discussions and informal conversations to reach individuals who were less comfortable attending formal sessions (Supplementary Materials 1 and 2). These interactions quickly revealed significant flaws in the initial engagement strategy and, more broadly, in the overall project approach. Community meetings highlighted concerns about the proposed seagrass planting. The primary concern was that increased biodiversity could trigger new conservation designations, potentially restricting boating, fishing and recreational activities. Residents also criticised the lack of early and inclusive consultation, noting that licence applications and small-scale trials had proceeded without broad community input. Additional concerns included limited familiarity with seagrass, potential impacts on boat engines, razor clam harvesting, shoreline debris and uncertainty over whether seagrass was historically present or being introduced as a "non-native" species. Despite these concerns, many members of the Dale and surrounding communities were supportive of the project, particularly in the context of seagrass as a nature-based solution to

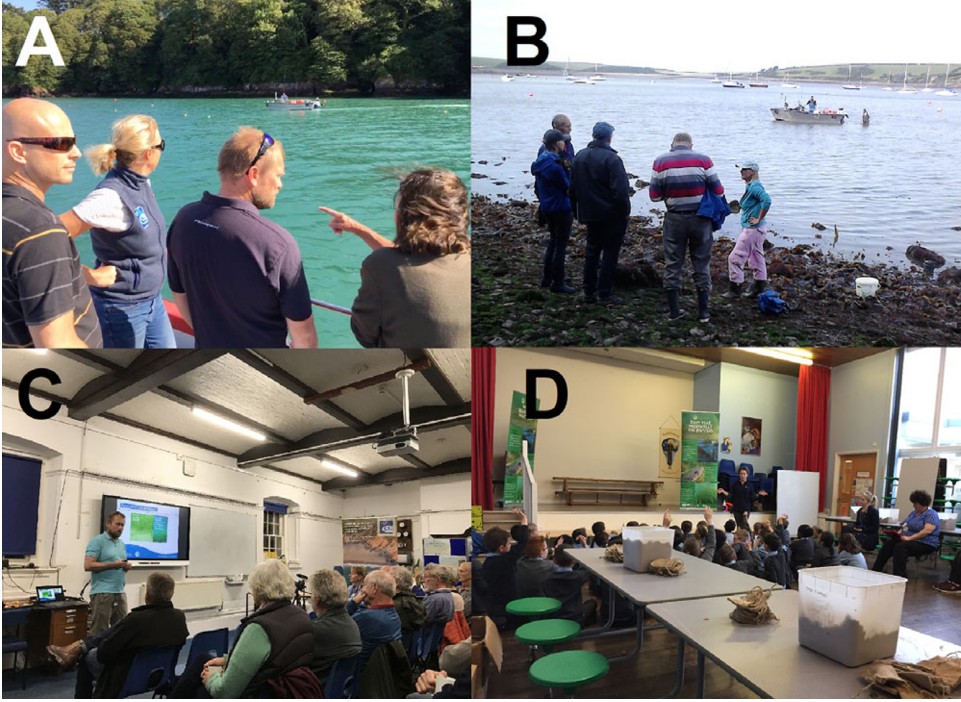

**Figure 1.** Engagement was conducted (a–d) throughout this project, which ultimately led to a coproduced project and eventually to more just restoration outcomes.

address the climate emergency and biodiversity crisis (Unsworth et al., 2022). In response to the range of community concerns, the project team paused development of the restoration project and conducted an online and paper-based survey to gain a clearer understanding of local perspectives on the proposed restoration (Engen et al., 2019) (see Supplementary Materials 1 and 2). The survey revealed general positive support for the project while highlighting the need to address specific concerns. Crucially, it also clarified the reasoning behind these concerns, providing actionable insights. Ultimately, the survey became a defining moment for the project, confirming and contextualising the views first expressed during stakeholder engagement meetings.

## Methods

### Co-production strategy

Following a positive response to the survey, a revised stakeholder engagement strategy was designed, adopting a "community co-production" approach. Co-production of knowledge has proven highly successful in terrestrial conservation (Nel et al., 2016; Schick et al., 2018) but is less common in marine settings (Páez et al., 2020), particularly in the context of marine restoration. Recent experience in Florida demonstrates that co-production with local stakeholders can enable projects of greater conservation ambition than those proposed by government regulators (Boucek et al., 2024).

### Activities

The restoration plan was co-produced through a series of regular public meetings and discussions with a group of local stakeholders (selected by the community), with strong input from wider members. This process led to adaptations to the project design (Gamborg et al., 2019), including restarting the licencing process at a higher government level with mandatory public consultation (a band 2 licence rather than band 1). Proposal details were then developed through a community-led workshop facilitated by a trusted, experienced, independent local facilitator, a role widely recognised as central to achieving mutually acceptable outcomes (van Maasakkers et al., 2014). The co-production process also included scrutiny of historical records, such as evidence of seagrass presence in Dale, and sought assurances from Swansea University, the legal licensee, regarding its ongoing responsibility for the project. Ongoing dialogue between the community and the project team resolved additional concerns, including an addendum to the Pembrokeshire SAC management scheme to protect boaters' rights to use Yacht Club anchorage areas. The project is also committed to investing in public-use moorings to provide an alternative to using the site as an anchorage and piloting a scheme to certify local fishers for their involvement in seagrass conservation. This led to the creation of the Project Seagrass "Sensitive Ecosystems Responsible Fisher" (SERF) Award (Unsworth, 2021), which remains active. The Welsh Government further provided assurances that they would not pursue previous proposals for a highly protected marine area in Dale. A successful community workshop (Figure 1) outlined alternative project proposals, beginning with a "back-to-basics" presentation on seagrass and the restoration objectives, which addressed concerns raised in the survey. To demonstrate that seagrass can coexist with active harbour use, community stakeholders visited Porthdinllaen in North Wales, where extensive seagrass meadows thrive despite significant boating and fishing activity. The lessons from that site visit were the integrated nature of site management across various stakeholders and the fact that it's an operational example of a seagrass social-ecological system in which biodiversity and human activity coexist.

### Outcomes

Outcomes of this community engagement included a revised project footprint, relocation away from existing moorings and fisheries storage pots, the provision of new public-use moorings and adjustments to BoSSLine deployment methods to reduce boat snagging risks (Yang et al., 2022). This new site was more exposed due to the prevailing wind direction from the south and west, bringing swell into the Milford Haven Waterway, yet it benefited from stronger community support and was still included within suitable bounds according to the habitat suitability model (Yang et al., 2022). The final licence application submitted to Natural Resources Wales incorporated extensive evidence addressing all concerns raised during community engagement, culminating in the licence being granted in February 2020.

## Results

### Project delivery

In August 2019, seed-laden shoots were collected from an intertidal-to-subtidal seagrass meadow in North Wales (proven successful in previous trials and of similar genetic provenance), with the aim of planting them in Dale Bay between October and November 2019. Due to the change in the engagement strategy, these shoots were stored over the winter of 2019. In February 2020, seeds were planted in Dale Bay within BoSSLines, using inert "play-sand" as the matrix for the hessian bags. Approximately 40 seeds were placed per bag, with 20,000 bags deployed along 20 km of sisal rope Figure 2. A further 450,000 seeds were planted in November 2020 (Unsworth et al., 2024), including 160,000 seeds in an experimental setup to evaluate the effectiveness of restoration methods. Despite the earlier disagreements and challenges, the initial planting facilitated interactions between project staff and community members, illustrating the trust developed through collaborative engagement.

### Volunteers and engagement

Following the successful engagement strategy and co-production of the project, a significant number of local people were involved in the planting process. In its first year, the project engaged 2,107 volunteers, contributing a total of 4,170 person-hours. Volunteers collected, processed and planted 800,000 seagrass seeds, working across organisations and logging 4,327 min underwater using SCUBA. Many volunteers were local to Dale and, empowered by the stakeholder agreements, actively contributed to the project. This approach not only facilitated project delivery but also engaged broader groups in marine conservation, enhancing public interest in seagrass ecosystems (Alamenciak and Murphy, 2024). Involving stakeholders in project delivery, alongside transparent reporting of data and findings, was recognised as essential for maintaining trust and building social capital (Pretty, 2003).

The project also resulted in extensive coverage of seagrass in the UK media, which was then amplified globally, reaching a wide range of stakeholders interested in seagrass conservation (Boiral and Heras-Saizarbitoria, 2017). Online reporting reached 174 million people, with a further 18.2 million reached through print media. Broadcast coverage included 31 pieces, alongside 12 online articles and 1 national print feature. The WWF later estimated the publicity value at £441,000. Given the limited local and global

recognition of seagrass and its ecological importance, this coverage improved the visibility of a habitat often referred to as the "*Ugly Duckling of Marine Conservation*" (Duarte et al., 2008).

## Discussion

### Ecological monitoring

Regular, transparent communication of results and project progress was essential for maintaining stakeholder interest and support during and after planting (Gornish et al., 2024). Initial snorkel and later SCUBA-based surveys were conducted in summer 2020, alongside the collection of additional seeds for autumn planting. Early observations indicated that 22% of bags assessed (from a 4% subsample) showed emerging shoots, averaging 2.3 ± 1.6 standard deviation shoots per bag, corresponding to an initial seed emergence rate of ~1.25% (Figure 3). By September 2020, seagrass density averaged 0.07 ± 0.09 shoots·m$^{-2}$, likely reflecting a decline from the initial emergence surveys (Figure 3a). Following further seed planting in autumn 2020, density increased to 0.38 ± 0.31 shoots·m$^{-2}$ by September 2021, indicating promising progress. Throughout this period, regular updates were shared with stakeholders via local newspapers and email, reinforcing knowledge exchange and community engagement (Gornish et al., 2024).

### Navigating incidents

In February 2021, a severe storm coincided with a trawling incident by a fisher not associated with Dale or the SERF Award scheme. The incident was documented by community members, reflecting the trust built during the project. Although the impacts of the storm and trawling cannot be completely separated, seagrass density dropped sharply to 0.04 ± 0.02 shoots·m$^{-2}$ in September 2021. Comparison with a similar storm in February 2019, which had minimal effect on seedlings, suggests that the trawling event was the primary cause. In

response, additional protections, including signage buoys, were installed to deter trawling, and conversations with the fishing community were undertaken. Small-scale supplementary plantings (~50,000 seeds between 2022 and 2024) were carried out.

As of October 2025, seagrass density had recovered to 0.75 ± 0.22 shoots·m$^{-2}$, with clumps reaching an average of 41.8 ± 6.2 shoots per clump, the highest recorded to date. Leaf lengths averaged 46.3 ± 21 cm, comparable to nearby reference sites (Bertelli et al., 2021). While the restoration footprint remains patchy, with large bare areas, community-led intertidal observations usingSeagrassSpotter. org revealed significant spread beyond the original restoration site (Figure 3b). This expansion likely results from a combination of self-seeding within the oldest patches and dispersion of seeds during planting, particularly those not enclosed in hessian bags.

Although many small-scale patches now achieve densities similar to local reference meadows (Bertelli et al., 2021), meadow-scale density remains low, with continuous coverage at <1% of a mature meadow (Figure 3). While the trajectory is positive, achieving a thick, continuous meadow at a landscape scale remains an ongoing challenge.

### Ongoing engagement

To ensure the project remained aligned with its original commitments, maintained a strong flow of information and provided a forum for ongoing discussion between the community and the project team, a formal Dale Seagrass Stakeholder Group was established. This group includes representatives from all major local organisations, as well as members of the community, functioning as a form of "co-management" for the future of seagrass at the site (Yandle, 2003). The group aims to meet every 6 months, and in December 2023 held a public presentation on the data collected to date. The formation of this group has enabled the local community to become embedded in the long-term stewardship of the seagrass in Dale, as the group makes decisions around future planting

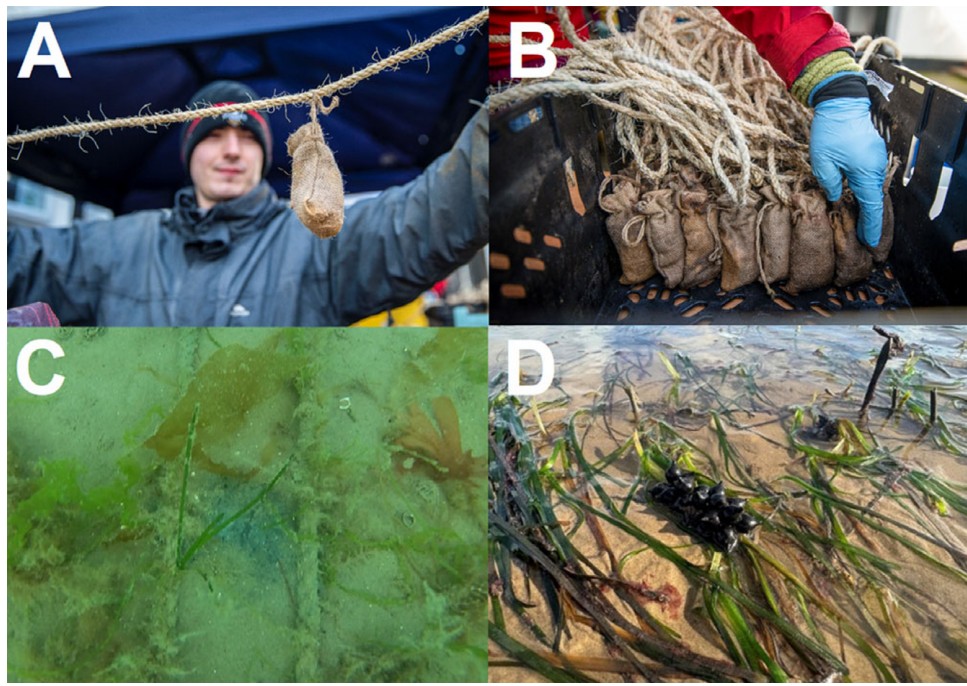

**Figure 2.** Hessian bag BOSSLines method being used in Dale for planting seagrass seeds (a and b), with seedlings appearing (c) and mature seagrass now present in Dale that supports cuttlefish eggs (d). Local volunteers were involved in all parts of the project delivery, helping to improve social capital.

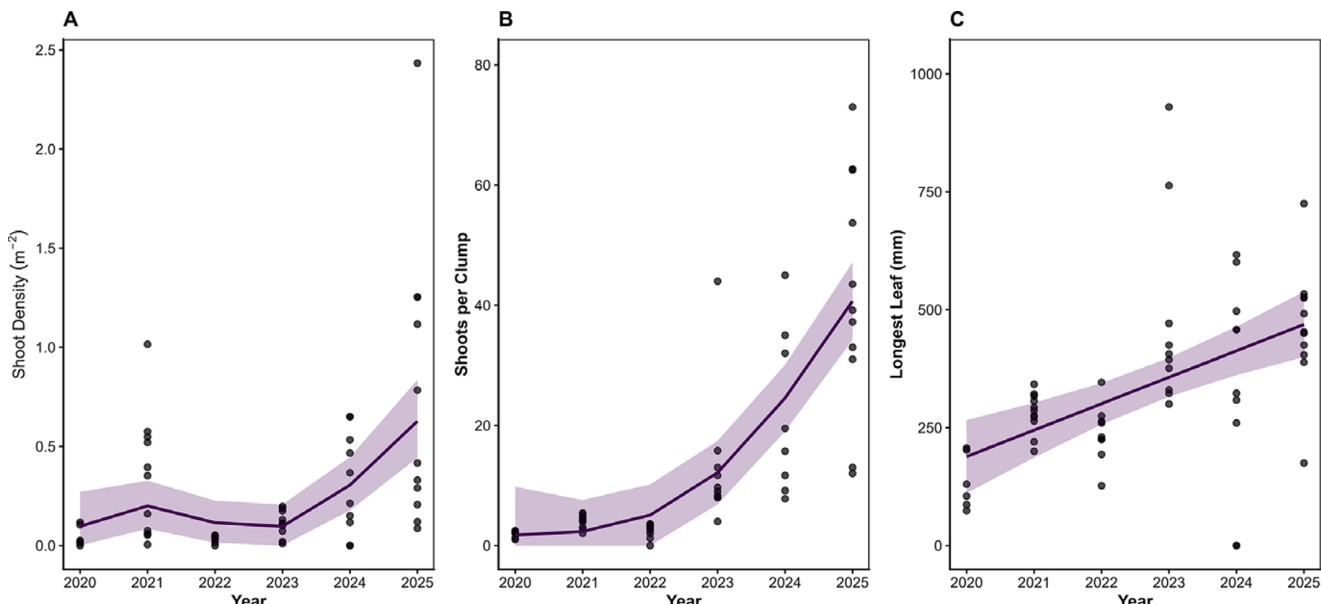

**Figure 3.** Seagrass shoot density, longest leaf and clump size have been monitored annually (2020–2025) over a series of 10 × 100 m long diver transects within the 2-ha seagrass restoration area in Dale, West Wales. Each panel displays observed values (points), the fitted trend from a generalised additive mixed model (solid line) and associated 95% confidence intervals (shaded ribbons). Panel a presents shoot density, Panel b shows the number of shoots per clump and Panel c illustrates maximum leaf length (mm). In all panels, individual observations are jittered or semi-transparent to illustrate sampling density, and GAMM smooths represent the best-fitting temporal trend while accounting for transect-level random effects.

locations, promotes awareness among the community and manages the seagrass moorings. Over time the group has also become involved with further elements of seagrass citizen science at the site to improve ecological monitoring.

Alongside the Stakeholder Group, a series of ongoing engagement activities has been developed from the initial engagement strategy, that continues to strengthen community support and foster a sense of ownership, creating a positive feedback loop in which trust enhances project outcomes and long-term sustainability (Figure 4). For instance, a local fisher was able to identify a gear alteration that could reduce their impact on seagrass. This gear alteration has now been made and trialled, resulting in reduced impact to seagrass from entanglement in shellfish pots but with no impact on catch reported by the local fisherman. The experience of this fisher has led to the expansion of conversations with fishers across South and West Wales, leading to local knowledge of seagrass helping to shape ongoing restoration. Incorporating fishers' experience and knowledge has been shown to contribute to better outcomes in marine initiatives (Gomes et al., 2025). Outdoor activity businesses are also facilitating educational sessions with young people, fostering ocean awareness and knowledge across generations (Ashley et al., 2019; Tian et al., 2023), as well as linking to the Welsh Government's Y Môr a Ni ocean literacy programme and the national species restoration programme Natur am Byth!

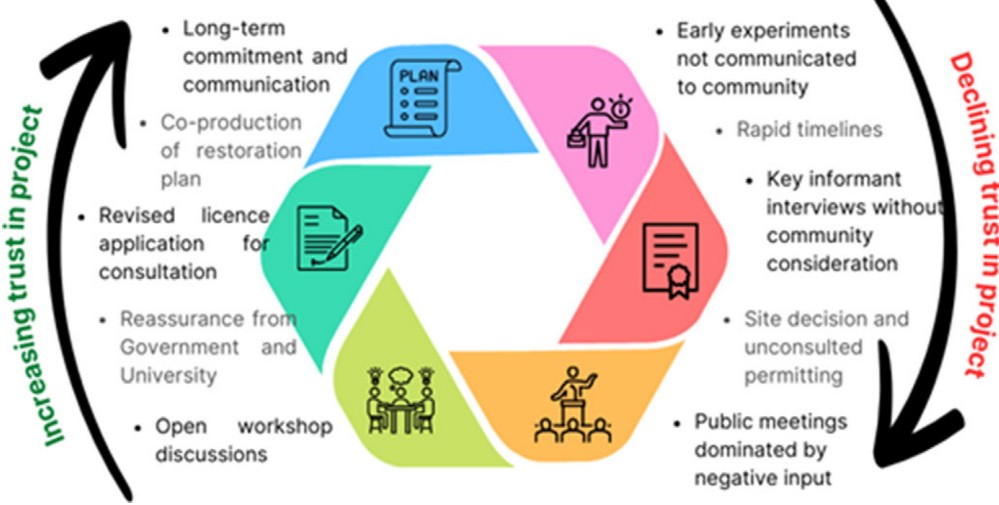

**Figure 4.** The Dale seagrass restoration project went through an initial process of challenging stakeholder engagement that reflected an inappropriate project design, creating a lack of trust between project proponents and the local community. Altering this design to one of co-production resulted in increasing trust and cooperation and facilitated the long-term success of the project.

**Table 1.** Recommendations from the local stakeholder engagement process (adapted from Burton and Germing, 2021)

| |
|---|
| 1. Stakeholder engagement benefits from pre-planning. |
| 2. Stakeholder engagement needs to start early so that time is available for appropriate and meaningful discussions and decision-making. |
| 3. Clearly demonstrate that stakeholder engagement is not a tick-box exercise and that stakeholders can coproduce project design. |
| 4. Stakeholder engagement needs to utilise multiple methods so that all stakeholders can be reached and feel comfortable inputting. |
| 5. Clear communication is essential to effective stakeholder engagement so that messages and discussions can remain factual and unbiased. This communication needs to be more widespread across all sectors of the community. |
| 6. Future statutory conservation reassurance was needed to attend to historical government actions. |
| 7. Sustaining the project long-term is necessary to ensure promises are delivered and local communities remain invested and protective. |

Local skippers are supporting annual dive monitoring, and free-diving groups are assisting with planting, all activities that will likely lead to long-term stewardship of the seagrass as local people are invested in its protection.

The community now view seagrass as a culturally and socially valuable asset. As one local resident noted: "The planted seagrass hasn't caused the negative change that many feared. In fact, people are proud that the village helped support such an important pioneering restoration project, and they really like the new visitor buoys as they support local tourism." The Dale experience – tracking the loss and subsequent recovery of community trust – has generated a series of recommendations on stakeholder engagement that may be valuable for others undertaking marine restoration projects (Table 1).

## Conclusions

This study reports on the development of a co-production model for seagrass restoration in West Wales, the first meadow-scale effort in the United Kingdom. Initial challenges stemmed from a stakeholder engagement strategy that did not involve all members of the community, leading to mistrust. A shift to a co-produced approach improved trust, cooperation and long-term prospects for success. While meadow-scale seagrass cover has not yet been achieved, patches now persist and are expanding beyond initial planting areas, showing signs of resilience. The original 2-ha target has not been met, but given that density and spread have not plateaued, 5 years is likely too short to judge overall success. The project has revealed critical lessons, particularly the importance of early and inclusive stakeholder engagement and ongoing, consistent monitoring. Ultimately, it demonstrates how co-produced ecological restoration can build resilient seagrass systems that benefit both people and nature.

**Open peer review.** To view the open peer review materials for this article, please visit http://doi.org/10.1017/cft.2025.10021.

**Supplementary material.** The supplementary material for this article can be found at http://doi.org/10.1017/cft.2025.10021.

**Data availability statement.** The data that support the findings of this study are available from the corresponding author, R.U., upon reasonable request.

**Author contribution.** RU conceived the research and wrote the draft manuscript. RU led field-based data collection. SB and JG developed the initial lessons-learned concept. RU, SB, RZ, JG, EF, BJ, LCU, FT, and CL edited the manuscript.

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
