## [Reviewer Report]

This manuscript provides clear and valuable insights into the use of co-production with local communities in seagrass restoration. It is well written, well structured, and supported by strong examples of engagement and adaptive management. The study makes a meaningful contribution to the literature on participatory approaches in coastal restoration.

Only minor revisions are needed to improve clarity and alignment. Some sections would benefit from more concise wording, clearer links between outcomes and supporting evidence, and closer consistency between the title, framing, and the seagrass-specific scope of the study. Addressing these points will enhance precision and overall readability, as mentioned below:

The current title “ ADVANCING MARINE RESTORATION THROUGH COMMUNITY

CO-PRODUCTION” refers broadly to marine restoration, but the manuscript focuses exclusively on seagrass restoration, both in terms of ecological context and the co-production process with local communities. The methods and conclusions focus solely on seagrass ecosystems. Keeping the title habitat-specific will ensure scientific accuracy and avoid overstating the generality of the findings.

The outcomes are relevant and clearly described (Lines 169-176), but the manuscript would benefit from clarifying the decision-making process, providing evidence for ecological suitability and explaining how the Porthdinllaen visit informed project design. Section “While the new site was more exposed to storms and potential commercial activity (e.g., trawling), it remained within ecological suitability bounds” (lines 172-173). Please include a brief description of the criteria or data used to make this statement. It would also strengthen the paper to discuss how these risks were assessed and managed. Lines 174-175: The visit to Porthdinllaen is a valuable example but could be expanded to clarify how lessons learned informed the final decisions. What lessons were transferred to the project?

Lines 254-260: The section provides good examples of continued community involvement; however, it would benefit from a clearer explanation of how these activities contribute to ecological outcomes, governance improvements, or long-term stewardship. I recommend linking the described activities (gear trials, education sessions, monitoring involvement) to specific benefits or expected impacts.

---

## [Reviewer Report]

This paper provides a good foundation representing co-production of restoration. As the authors state, it “describes a journey”, and is not presented in a systematic empirical fashion. I am a little surprised that such top-down restoration was proceeded with at the time, involving only a single stakeholder (yacht club). I would therefore have concerns about the ethical ramifications of this trial-and-error approach to co-production. Learning research lessons at the expense of communities is a dangerous approach, and one which should have been more thoroughly understood in 2019, when the project initiated. Lessons in the failure of top-down conservation approaches are far from new. However, the field has matured considerably since then, and the case study presented by this paper provides solid precedent for future co-creation, unfortunately by showing how ‘not to do’ engagement.

The organisation of this submission is a little haphazard, with section headings that are difficult to follow or that are too narrative in focus. Sticking to the Intro-Methods-Results-Discussion format may allow for easier reading. At present, it is a strong narrative but it reads a little journalistic.

Generally, it is unclear how this represents true co-production, as communities do not appear to have been involved in the design of restoration areas or techniques. Rather, lessons were learned from poor communications and this was improved upon. The volunteer numbers are fantastic, but this is more citizen science or engagement than co-production.

Details comments are included below:

- A quick search for double spaces would be helpful, as I spotted a few

- Line 74, what is meant by “anthropogenic risk”?

- I appreciate that this field is emergent, but the citation of the work of Unsworth is disproportionate in this paper. More effort is required to capture the work of other seagrass researchers elsewhere.

- Line 132: Can you please include the number of responses you received to the survey, and the demographics of these? On line 139, this is described as “overwhelmingly positive”. Please stick to objective language, unless there is evidence for being overwhelmed?

- Line 146: More subjective language – “can even…”

- Line 149: What is a “loosely structured group…”??

- Line 153: Confidentiality retained, but more context is required on the “respected local individual” and I don’t think the reference is relevant here.

- Mitigation to Manage Community Concerns: This section is presented as a highly descriptive passage of trial-and-error engagement, showing reactionary responses to stakeholder concerns. As a manuscript, this needs to be framed better as a methods / process section, rather than a verbatim description of what was done. Otherwise, it communicates as an ad hoc fire-fighting exercise.

- Line 181: collection in north Wales for use in south Wales. Any consideration of genetic provenance here and suitability thereof?

- Line 191: Wow, an impressive number of volunteers!

- Line 197: More subjective language: “remarkably”, “just”

- Line 199: Is there evidence of this “positive interaction”? Otherwise, a meaningless subjective statement. In the context of this paper, evidence to support the success of this process is essential.

- Line 207-210: I do not believe the media reporting is relevant to this paper.

- Line 250: Given the trawling incident, I would hope that fishing organisation are included in this group

- Line 332: Journal name is all-caps, which it shouldn’t be.

- Table 1: The first point is lacking a number

- Table 1: I would also love to see a deeper consideration of how “engagement” and “communication” can be improved, instead of just increasing the cadence of it.

- Captions: Fig. 1: More subjective language – “significant”.

- I dispute the fact these led to stronger restoration outcomes, as the outcomes were not strong. Maybe they would have been stronger by ignoring the community entirely?

---

## [Editor Report]

Thank-you for providing an interesting and relevant case study; we look forward to your revised manuscript addressing the comments of the two reviewers. A suggested change to the title could be “Advancing seagrass restoration through community engagement”

---

## [Editor Report]

Handling editor: 

The reviewers’ inputs are adequately addressed, and the manuscript is accepted for publication. Detail is provided where needed and revisions have been made to ensure a more objective narrative tone. The Methods section is now presented under headings on Co-production strategy, activities and outcomes.

Thank-you authors for your contribution on stakeholder engagement in marine restoration.